# Sorted stem/progenitor epithelial cells of pubertal bovine mammary gland present limited potential to reconstitute an organised mammary epithelium after transplantation

Laurence Finot[1], Cathy Hue-Beauvais[2¤], Etienne Aujean[2], Fabienne Le Provost[2], Eric Chanat[1]*

1 PEGASE, INRAE, Institut Agro, Saint Gilles, Paris, France, 2 Université Paris-Saclay, INRAE, AgroParisTech, GABI, Jouy-en-Josas, Paris, France

¤ Current address: SDAR, INRAE, Centre Ile de France Jouy-en-Josas, Antony, Paris, France
* eric.chanat@inrae.fr

**Data Availability Statement:** Original blot/gel images as well as all other images and data related

## Abstract

The development and maintenance of mammary gland tissue depend on the proliferation and differentiation of mammary stem and progenitor cells. Here, we investigated populations of mammary epithelial cells that are potential candidates for bovine mammary gland development using xenotransplantation into mice cleared mammary fat pad. Transplanted mammary explants from 17-month-old Holstein heifers developed outgrowths exhibiting the archetypal morphology and molecular marker distributions of the bovine gland. Xenotransplantation of sorted mammary epithelial cells (CD49$_f^{pos}$) into bovinised fat pads using inactivated bovine fibroblasts resulted in outgrowth developments with 50% take rate, but these lacked the ductal or alveolar epithelial structures of the normal mammary gland. Similar results were obtained with xenografts of candidate bovine mammary epithelial stem cells (CD49$_f^{high}$CD24$^{pos}$) or epithelial cells of the basal lineage (CD49$_f^{high}$CD24$^{neg}$) which also developed as clumps of cells surrounded by stromal stretches within the mouse adipose tissue. In conclusion, sorted cells showed compromised regenerative potential for epithelial morphogenesis. Further work is therefore needed to identify mammary stem/progenitor cells with full regenerative capabilities for biogenesis of normal mammary gland structure, with milk-secreting function.

## Introduction

A major goal in the field of mammary gland biology/physiology is to understand how the mammary gland structure develops in order to secrete huge quantity of milk. One research route to this end is to identify the mammary stem cells (MaSC) and progenitor cells that are involved in the building of the mammary gland tissue, especially those that give birth to mammary epithelial cells. With this aim, we previously characterised different types of epithelial

to this study are publicly available from the INRAE repository (https://doi.org/10.57745/NQZKHI).

**Funding:** The author(s) received no specific funding for this work.

cells involved in mammogenesis, as well as their phenotypic evolution and molecular signature at key physiological stages (puberty, lactation and drying off) in dairy cows [1].

Inspired by the literature on the lineage of mammary epithelial cells in bovine [2] but also in mice [3] and human [4], our approach to identify bovine MaSC focused on the expression of cell surface antigens, or cluster of differentiations (CD), that were demonstrated to be common among these species, namely $CD49_f$ (alpha-6 integrin), a marker of mammary stemness [3, 5], and CD24 (signal transducer CD24) [6]. We highlighted four epithelial cell subpopulations including two subpopulations with high level of $CD49_f$ one displaying CD24 protein expression [1]. Based on previous reports showing that, in bovine, the $CD49_f^{high} CD24^{low/med}$ cell population was enriched in cells that are at the top of the cell hierarchy [2, 7], we anticipated that the subpopulations we characterised contained the MaSC and progenitor cells, respectively. This conclusion was supported by the observation that mice mammary epithelial cells sorted according to CD24 expression, and high protein expression of $CD49_f$, were in capacity to reconstitute a functional mammary tissue *in vivo* upon mammary transplantation in mice [3, 8]. As a result, we further established the identity of these two bovine epithelial cell subpopulations based on their molecular signature (gene expression and cellular phenotype) and *in vitro* assays [9].

The *in vitro* assays used in the stem cell research field were long considered as a surrogate to functional *in vivo* assessment. The ability of cells to form spheres (mammospheres in the present case) when cultured in 3D within hydrogel (e.g., Matrigel™), or in suspension, is attributed to both resistance to anoikis and proliferative capacity, two stemness properties [2, 10]. Although these assays are informative and easy to implement, they clearly show their limits when it comes to fully highlighting the expected proliferation and differentiation capabilities of activated MaSC. In contrast to these functional *in vitro* assays, transplantation in the mammary gland makes it possible to preserve the local microenvironment essential for the full development of MaSC. In conclusion, when the objective is to prove the identity of a putative MaSC population by demonstrating their ability to regenerate the organisation of mammary tissue, mammary transplantation is a standard functional test.

The mammary transplantation assay has proven essential for the assessment of stem cells endowed with the capacity of self-renew and differentiation, and to highlight their capacity to reconstitute epithelial tissue in the cleared mammary fat pad of recipient mice. Since the pioneering study by the DeOme group who developed an epithelium-cleared mammary fat pad technique in 1959 [11], the mammary transplantation assay constantly evolved. Initially used exclusively to transplant mouse tissue, the technique was gradually adapted to transplant non-murine samples into an orthotopic xenograft model, or xenotransplantation assay, to test human tissues. Nevertheless, researches on human breast stem and stem-like cells using this technique encountered challenges due to the failure of human breast epithelial cells to colonize mouse mammary fat pads. However, mammary tissue transplantation has been shown to be effective for both human and bovine species [12, 13]. Indeed, in their pioneering bovine mammary tissue transplantation experiment, Sheffield and Welsh showed that grafted tissue slices maintained normal mammary gland morphology and responded to mammogenic hormonal growth stimulus upon injection of oestradiol, progesterone, growth hormone and prolactin [13]. These data suggested that it is the xenotransplantation of isolated cells that causes developmental problems. Importantly, it was then demonstrated that species specific differences in terms of structure and composition of the mammary gland between mice and human, or bovine, hampered the extensive development of the foreign epithelium in the recipient mouse, and therefore the assessment of the developmental capacities of the dissociated sorted cells, such as human or bovine epithelial cells and MaSC [14]. Human breast, like the bovine mammary gland, contains much more fibrous connective tissue surrounding the epithelial

structures (ducts and alveoli) than encountered in mice where the stroma is mainly adipose-rich. Proia and Kuperwasser took this distinct stromal composition into consideration and successfully achieved human graft development by humanizing the murine recipient mammary fat pad by pre or co-implanting human fibroblasts [15], thereby providing within the recipient mammary gland a supportive fibrous microenvironment in which human mammary organoids or cells could successfully grow. Humanization of the mouse mammary gland relied on the injection of irradiated cells prepared from a human mammary stromal cell line. In the group of Kuperwasser, the most successful humanization approach consisted in pre-implanting irradiated immortalized human breast fibroblasts before transplanting xenografts containing a mix of candidate mammoplastic organoids with primary human fibroblasts. This enabled the production of human hyperplastic and neoplastic outgrowths in mice mammary fat pads [16]. An analogous xenotransplantation approach using irradiated fibroblasts from the murine cell line 10T1/2 was developed for attempting to demonstrate that a subset of primary bovine cells retained regenerative capabilities after a prolonged period of culture *in vitro* [17]. Interestingly, the functional capacity of these cells was associated to a stem-like phenotype characterised by a significant expression of $CD49_f$ [7].

In our search for identifying the cells at the top of the bovine mammary cell hierarchy, we adopted the functional xenotransplantation approach to verify the regenerative potential of selected populations. Our main objective was to evaluate the *in vivo* developmental and differentiation capacities of the bovine epithelial cell populations we presume to be stem and progenitor cells. These putative bovine MaSC and progenitor cells were sorted on the basis of $CD49_f$ and CD24 expression: mammary epithelial cells ($CD49_f^{pos}$), mammary epithelial stem cells ($CD49_f^{high} CD24^{pos}$) and $CD49_f^{high} CD24^{neg}$ cells. With regard to the peculiar fibrous composition of the bovine mammary gland, we choose to implant bovine fibroblasts to bovinise the recipient mice cleared fat pad to optimize conditions supporting growth and differentiation of those bovine MaSC.

## Material and methods

### Animals

**Ethical statement.** Mice experiments were performed at the INRAE experimental facility (Unité expérimentale 0907 infectiologie expérimentale des rongeurs et poissons, Domaine de Vilvert, Jouy-en-Josas, France; https://doi.org/10.15454/1.5572427140471238E12) in compliance with the ethical standards of the European Community (Directive 2010/63/EU) and were approved by the regional ethical committee (Comité d'Ethique en matière d'expérimentation animale or COMETHEA, approval numbers APAFIS#7332–2016060214463382v3 and APAFIS#14945-2018050415489992v3).

**Mice.** Immunodeficient female mice (BALB/c AnNRJ-Foxn1 nu/nu) were purchased from Janvier Labs (St Berthevin, France). Mice were housed in group under specific-pathogen-free and controlled environmental conditions (22–24°C, 12 h light/dark cycle and 50 ± 10% humidity), with free access to food and water. Mice were euthanized via cervical dislocation.

**Bovine.** The mammary gland parenchyma and the mammary subcutaneous adipose tissue used to produce mammary xenografts or to isolate fibroblasts were obtained from three pubertal Holstein heifers [9] housed at the INRAE experimental dairy farm of Méjusseaume (IE PL, INRAE, 2021. Dairy nutrition and physiology, https://doi.org/10.15454/yk9q-pf68) and sacrificed at 17 months of age at the slaughterhouse of Gallais Viande (Montauban-de-Bretagne, France) following standard commercial practices.

## Preparation of donor transplant

For the preparation of bovine xenografts (mammary explants, dissociated single cells or sorted subpopulations), the mammary glands of the sacrificed 17-month-old heifers were collected and mammary gland pieces were excised and sampled immediately into small explants ($\approx$3 mm$^3$). Explants were suspended in 90% foetal bovine serum (Gibco Invitrogen Saint Aubin, France)/ 10% dimethyl sulfoxide (Sigma-Aldrich, Saint-Quentin Fallavier, France), slowly frozen in cryovials at -80˚C, and stored at -150˚C until to their use.

**Mammary explant xenograft.** Murine mammary explants used as positive controls for transplantation were extemporaneously prepared from mature female FVB/N wild-type mice at mid-gestation. Animals were euthanised via cervical dislocation and mammary inguinal gland pieces were collected. The mammary parenchyma was washed with sterile Phosphate Buffered Saline (PBS) containing 1% penicillin streptomycin mix (Fisher scientific, Illkirch-Graffenstaden, France) and cut into small explants ($\approx$1 mm$^3$). To prepare bovine mammary explants, cryo-conserved bovine mammary explants were rapidly thawed at 37˚C, washed in sterile PBS (Fisher Scientific), and further cut into $\approx$1 mm$^3$ explants.

**Sorted cells from mammary dissociated explant.** Cryo-conserved explants from bovine mammary parenchyma were rapidly thawed at 37˚C, washed in sterile PBS and enzymatically dissociated as previously described [9] to generate a single cell suspension. Dissociated cells were labelled with the anti-CD49$_f$ antibody (see S1 Table) and bovine mammary epithelial cells (bMEC) were sorted as CD49$_f^{pos}$ cells. These cells were engrafted either three weeks after or together with bovine fibroblasts (see below). The two candidate mammary stem cell populations were sorted on the basis of the expression of CD49$_f$ and CD24, CD49$_f^{high}$CD24$^{neg}$ and CD49$_f^{high}$CD24$^{pos}$ cell subpopulations, following labelling with both anti-CD49$_f$ and anti-CD24 antibodies. Briefly, cells were incubated with the relevant antibodies for 20 min at 4˚C, washed and re-suspended in MACSQuant Running Buffer (Miltenyi Biotec, Paris, France) supplemented with 0.5% MACS® bovine serum albumin Stock Solution (Miltenyi Biotec). Labelled cells were sorted using a BD FACS ARIA II flow cytometer (BIOSIT CytomeTRI technical platform, Villejean Campus, Rennes, France). Cells were sorted in the 0-16-0 sort precision mode with 70 μm nozzle. This setting was shown to recover sorted cells close to 100% purity [18]. Sorted cells were centrifuged at 300 G for 5 min at 4˚C and stored in 90% foetal bovine serum/10% dimethyl sulfoxide at -150˚C until transplantation. Viability of cryo-conserved cells was evaluated using trypan blue (BIO-RAD, Marnes-La-Coquette, France) using the automated cell counter TC20 (BIO-RAD). At the time of transplantation, sorted cells were rapidly thawed at 37˚C, washed with DMEM (Dulbecco's modified Eagle's medium; Fisher scientific) and counted. Sorted cells (15,000 cells for bMEC or 5,000 cells for the putative stem/progenitor populations) were mixed with 1.5 10$^5$ gamma-irradiated bovine fibroblasts (see below) in a final volume of ten microliters and injected into cleared fat pads using a 26-gauge needle mounted on a 50 μl Hamilton glass syringe (Co-transplantation). In some instances, gamma-irradiated bovine fibroblasts were transplanted into cleared fat pads three weeks before bMEC (Pre-transplantation).

**Production and characterisation of bovine fibroblasts.** Bovine fibroblasts were isolated from subcutaneous adipose tissue of 17-month-old heifers as previously described [19] with some modifications. Briefly, subcutaneous tissue was minced and digested in 2 ml per g of tissue of Krebs-Ringer-bicarbonate (Sigma-Aldrich) buffered with 10 mM HEPES (Fisher Scientific), supplemented with 3% bovine serum albumin (Roche Diagnostics, Meylan, France) and 1.3 mg/ml collagenase A (Merck, St Quentin Fallavier, France), in a shaking water bath for 60 min at 37˚C. The fraction of stromal cells, which includes bovine fibroblasts, was separated from floating adipocytes by centrifugations at 800 G for 10 min followed by 400 G for 10 min, all at room temperature. Pelleted cells were resuspended in DMEM (Fisher scientific), filtered

through 200-, then 25-μm nylon mesh and washed twice with PBS. Cells were resuspended in PBS and bovine fibroblasts were recovered by pre-plating the cell suspension in 25 cm$^2$ T-flask for 1 h in a humidified incubator at 37˚C and 5% CO2. Adherent cells, referred to as bovine fibroblasts, were washed three times with PBS to remove non-adherent cells. Bovine fibroblasts were then amplified and grown in fibroblast growth medium composed of DMEM supplemented with 10% foetal bovine serum (Fisher scientific), 5 μg/ml hydrocortisone (Merck) and 1% penicillin streptomycin mix (Fisher scientific). Primary bovine fibroblasts were maintained in culture during 15 passages before their first use in transplantation and up to passage 20 to ensure the homogeneity of the cell cultures. Bovine fibroblasts were passed when they reached confluency by trypsinisation with 0.25% trypsin–ethylenediaminetetraacetic acid solution (Fisher scientific) and subcultured at 1:10 dilution.

To assess the typology of the cells and the purity of the bovine fibroblasts culture (see S2 Fig), immunofluorescence assays were performed at passages five, ten and fifteen. We measured the viability of bovine fibroblasts during passaging by trypan blue labelling and counting using the automated cell counter TC20. For determination of the doubling time, bovine fibroblasts were plated at a density of 100,000 cells in 6-well plate and cultured for 48 h before being trypsinised, labelled with trypan blue and counted as above. The expression of mesenchymal proteins and the absence of epithelial markers were analysed at passage fifteen, i.e., before their use for transplantation, by both western blotting and immunofluorescence. Impairment of cell proliferation of irradiated bovine fibroblast was tested by PCNA immunoblotting.

**Irradiation of bovine fibroblasts.** Bovine fibroblasts were expanded in 75 cm$^2$ T-flasks to confluency and X-ray irradiated at 50 Gy (1.41 Gy/min) to block proliferation. Between irradiation and transplantation, fibroblasts were maintained in culture in fresh fibroblast growth medium for a maximum of two days.

## Mice surgery

Three-week-old host mice were injected intra-peritoneally with 200 μl of the analgesic Finadyn (0.2 mg/ml) and anesthetised with isofluorane (2.5% L/min). Mice were then kept with a face-mask supplying isofluorane and oxygen during the whole procedure. A mid-sagittal incision through the skin was made across the pelvis and toward each leg to expose the two 4th inguinal mammary glands. The fat pads were cleared of their endogenous epithelium by removing both the mammary tissue anterior to the lymph node and the lymph node after cauterisation of the blood vessels with an electric scalpel [11]. The surgical procedure lasted around 10 min and the total duration of general analgesia was ≈ 20 min. Donor transplants, including bovine or mice explants, sorted cells with bovine fibroblasts (co-transplantation), or bovine fibroblasts alone (pre-transplantation), were transplanted bilaterally into the 4th cleared mammary fat pads. In the case of pre-transplantations, transplantations of mammary bovine samples were performed three weeks after fat pad clearing and bovine fibroblasts injection. The surgically incisions were closed with wound staples. Animals were removed from the isofluorane stream, place on a warming pad, and monitored until they were awake and mobile. Mice were further monitored during the following week for any sign of pain, infection or surgical complications. To ascertain the removal of the entire rudimentary endogenous epithelium during the preparation of cleared mammary fat pad, excised tissue pieces were spread on a microscope glass slide and stained with Carmine Alum (see whole mount below).

## Xenograft assays

Ten weeks post-transplantation, mice were euthanised to collect the mammary glands for proceeding to whole mount staining and immunohistological analysis.

**Whole mount.** Whole mounts were carried out as previously published [20]. Briefly, mammary glands were spread on glass slides, fixed in Carnoy's solution (ethanol/chloroform/glacial acetic acid 6/3/1) overnight at room temperature, and rehydrated gradually. The tissue was stained with Carmine Alum solution for 1h30 at 4˚C, dehydrated, cleared in xylene (Merck) and mounted using Permount™ mounting medium. Images of the stained mammary glands were acquired using a LEICA M80 binocular magnifier equipped with a LEICA MC170 HD camera.

**Immunohistochemical assays.** Outgrowths were dissected from Carmine Alum-stained whole mounts and processed for inclusion in paraffin blocks. For bovine mammary gland tissue analysis, samples were washed in PBS, fixed in 4% paraformaldehyde for 2h, embedded in paraffin. Paraffin sections (5 μm thick) were rehydrated and stained with haematoxylin and eosin solution. Digital images were acquired using a Hamamatsu NanoZoomer (Hamamatsu Photonics, Tokyo, Japan).

Immunostaining was performed on paraffin-embedded xenograft sections as described in Finot et al., 2018 [9]. Briefly, deparaffinised sections were incubated with 50mM ammonium chloride (Merck) for 10 min and with 0.1% Sudan black B (Merck) in 70% ethanol for 20 min to quench the autofluorescence of immune cells. Slides were rinsed with Tris-buffered saline containing 0.02% Tween-20 (Merck) and tissue sections were subjected to heat-induced epitope retrieval in 1mM ethylenediaminetetraacetic acid (Merck), pH8, using a microwave at 800 watts for two periods of 5 min. For the labelling of Ki67, epitope retrieval was performed using 1X ImmunoDNA retriever citrate buffer (Diagomics) and the low-pressure pre-program (106–110˚C during 15 min) with the Bio SB TintoRetriever Pressure Cooker (Diagomics, Blagnac, France). The protocol for the following steps of permeabilisation, blocking and immunostaining is detailed below (see section immunofluorescence).

## Miscellaneous analytical methods

**Western blotting.** Proteins from bovine fibroblasts and mammary parenchyma were extracted, quantified and analysed by Western blotting as previously described [21]. Briefly, proteins were extracted from pelleted fibroblasts using the RIPA extraction reagent (Fisher Scientific) or from mammary tissue using the T-PER tissue protein extraction reagent (Fisher Scientific). After incubation and centrifugation at 13,000 G for 10 min at 4˚C, the soluble protein fraction was collected and quantified using the BCA assay kit (Fisher Scientific). For each sample, 10 μg of protein were resolved by SDS-PAGE and electrotransferred to PVDF membranes. Incubation with primary antibodies (see S1 Table) was for overnight at 4˚C. Membranes were incubated with relevant secondary antibodies for 2 h at room temperature and immunoreactive proteins were revealed using ECL substrate (Fisher Scientific). Light signal was digitalised using the ImageQuant LAS4000 Imager digital system (GE Healthcare, Velizy-Villacoublay, France) and quantified with the ImageQuant TL software (GE Healthcare).

**Immunofluorescence.** The immunofluorescence assay was essentially as previously described [9]. Bovine fibroblasts were grown on glass coverslips in 6-well plates and fixed for 30 min with 4% paraformaldehyde at room temperature. Cells were incubated with 50mM ammonium chloride for 10 min and permeabilised with 0.25% Triton X-100 (Merck) for 5 min. Nonspecific antibody blocking was for 1 h with 2% bovine serum albumin (Merck) in Tris-buffered saline. Cells were incubated for 2 h with primary antibodies and for 45 min with secondary antibodies (see S1 Table) at 37˚C. After washing, nuclei were counterstained with bisBenzimide (Hoechst) H 33342 (Merck) at 1 μg/mL for 2 min. Glass coverslips were mounted using Vectashield mounting medium (Vector Laboratories, Burlingame, CA). Images were captured with an Apotome™ and the Zen software (Zeiss France).

For Ki67 quantification, images of murine or bovine outgrowths and bovine explants before transplantation were analysed with ImageJ software (W. Rasband, National Institutes of Health, Bethesda, MD). Five images were processed for each samples using the following parameters: scale set in pixels, split of colour channels (Ki67 channels and nuclei), threshold images adjusted to 50 ± 5, watershed of the binary process and particle analysis (size from 100 to infinity and circularity from 0.1 to 1). The percentages of proliferating cells were calculated as the ratio of the number of Ki67-labelled nuclei to the total number of Hoechst-labelled nuclei.

## Results

### Xenografts of heifer mammary gland explant regenerate a bovine-like mammary epithelium

To study the stem and progenitor cell populations of the bovine mammary gland, we developed a xenotransplantation assay in the cleared mammary fat pad of immunocompromised mice. With this aim, we first transplanted mammary explants from pubertal cows and mouse mammary gland explants as controls (Fig 1). Mice mammary glands were collected ten weeks later and the development of the xenograft was evaluated using whole mount analysis. As shown in Fig 1, transplantation of mouse mammary gland explants gave rise to the development of outgrowths with the characteristic ductal tree of mouse mammary epithelium (Fig 1A, top, for comparison with virgin mouse tissue see S1 Fig). The morphology of the outgrowths resulting from the transplantation of mammary gland explants from heifer was completely different (Fig 1A, bottom, dotted black line). Outgrowths showed a compact appearance and did not spread through the entire fat pad, as does the neo-formed murine epithelium. However, bovine explant outgrowths were observed following 75% of the transplantations, as compared with 100% when the xenografts were from mouse (Fig 1D). Given the present observation, it was important to further characterise these bovine-derived outgrowths. First, morphological observation using haematoxylin and eosin-stained histological section revealed the development of a well-defined ductal or alveolar epithelium within stroma, surrounding open lumens (Fig 1B). Most, if not all, epithelium figures were bi- or multi-layered (Fig 1B, inset). The outgrowths were also vascularised (Fig 1B, black arrows). On a general note, the morphology of these outgrowth developments was highly reminiscent of bovine mammary gland epithelium at puberty (see S1 Fig and Fig 1 in [1]). The bovine nature of such outgrowths was also confirmed in a previous study by PCR amplification of genomic DNA after extraction from histological samples [20]. To further characterise the cell populations composing this mammary parenchyma, we analysed the distribution of epithelial cell markers by immunofluorescence. As shown in Fig 1C (Top panel), the luminal epithelial cell marker cytokeratin 7 was mostly found in close apposition to the lumens, staining the apical moieties of the epithelial cell surrounding the lumens. Although weaker, staining was also observed between cells, at their lateral membrane surfaces. An identical labelling was observed with the luminal epithelial cell marker cytokeratin 19 (Fig 1C, middle panel). The cells surrounding these epithelial structures formed a continuous monolayer that was strongly labelled by the basal marker cytokeratin 14 (Fig 1C, top panel). A particularly fibrous stroma was observed between the epithelial structures and it should be noted that its appearance is reminiscent to that found in the bovine mammary gland. As expected, the stromal part of the tissue which contains few cells was only labelled by the stromal marker collagen type 1. Finally, we found that epithelial cells of the xenograft proliferate as demonstrated by labelling for Ki67 (Fig 1C, bottom panel). Quantification showed that the proliferation rate was ≈ 2,8% in the grafted bovine explant (Fig 1E).

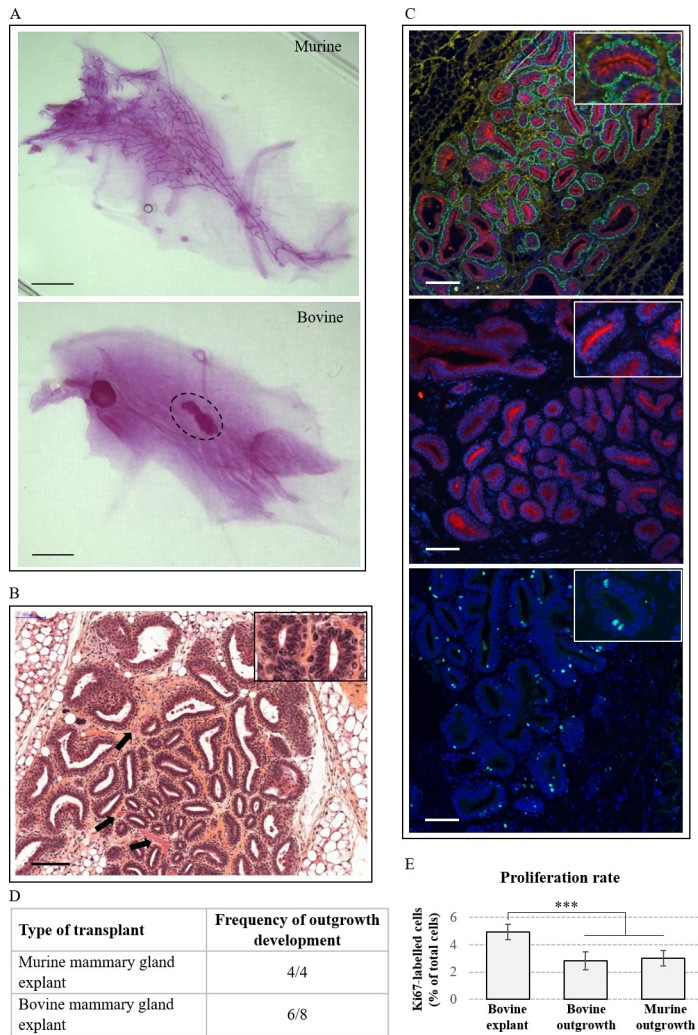

**Fig 1. Xenotransplantation of bovine mammary explant into the cleared mammary fat pad of mice results in bovine-like epithelial tissue outgrowths.** (A) Representative carmine-stained whole mount of mice mammary gland transplanted with either murine or bovine mammary gland explant. The dotted black line in the bottom panel surrounds the typical structure obtained with bovine xenografts. Scale bar = 2 mm. (B-C) Characterisation of the outgrowths obtained following transplantation with bovine mammary gland explants. A four times magnification of an epithelial structure is shown in the top right insets. (B) Hematoxylin and eosin-stained histological section of a representative outgrowth shows epithelial developments within stroma. Most lumens appeared surrounded by at least two layers of cells. Some blood vessels are indicated by arrows. Scale bar = 100 μm. (C) The typology of the cells in outgrowths was determined by immunofluorescence staining of histological sections. Top panel: basal marker cytokeratin 14 (green), luminal marker cytokeratin 7 (red) and stromal marker collagen type I (yellow). Middle panel: luminal marker cytokeratin 19 (red). Bottom panel: proliferation marker Ki67. Nuclei were counterstained with Hoechst 33342 (blue). Scale bar = 100 μm. (D) Frequency of outgrowth development following transplantation with either murine or bovine mammary gland explants. (E) Frequency of Ki67 labelled cells in the indicated tissue samples. ***$P \leq 0,0001$.

Interestingly, it was quite similar in the grafted murine explant with 3.2% of Ki67-labelled cells. Although lower, these rates of proliferation were in the same order of magnitude than that found in the bovine explant before transplantation (4.8%). We concluded from these results that bovine-like mammary gland tissue develops following xenotransplantation of mammary gland explant from heifer.

## Xenograft of sorted bovine mammary epithelial cells into bovinised mammary fat pad leads to tissue outgrowth developments

In order to further identify and characterise the cells responsible for the development of the mammary gland epithelium observed in the experiments above, we tested the transplantation of bovine mammary epithelial cells (bMEC). In line with this, we have demonstrated in a previous study that the cells oriented towards the epithelial development in the bovine mammary gland expressed the protein $CD49_f$ [9]. Since the bovine mammary gland has a more complex fibrous mammary stroma, as mentioned above, mouse cleared mammary fat pads were bovinised by injecting gamma-irradiated bovine fibroblasts to increase the chances of successful transplantation. For this purpose, we have established a clone of fibroblasts that was derived from primary culture of bovine fibroblasts that were isolated from heifer mammary subcutaneous adipose tissue and sub-cultured during fifteen passages before their first use in the xenotransplantation assay (S2 Fig). Indeed, in their pioneering report, Rauner and Barash transplanted mouse fibroblastic cells to facilitate the implantation of bovine mammary epithelial cells (Rauner and Barash, 2013). Then, they use the bovine cells remaining after the preparation of mammary epithelial cells (Rauner and Barash, 2016). Instead, we finally chose to develop a bovine fibroblastic cell line in order to standardize the xenotransplantation procedure. Moreover, in contrast to the above studies which involved the implantation of a 1:1 mix of native and irradiated cells, we decided to implant only irradiated fibroblast. Irradiation of the fibroblasts is required to block their proliferation in the mice mammary gland, a phenomenon that would be detrimental to the development of the relevant co-transplanted cells. Indeed, we thought that implantation of native bovine fibroblasts could be at risk, these being potentially in a position to simply develop within the fat pad or to undergo a mesenchymal to epithelial transition, both phenomena potentially making it difficult to interpret the observed tissue developments, the fibroblast-derived tissue being of the same species than the injected sorted mammary epithelial cells. To ensure that gamma irradiation stops the proliferation of fibroblasts but has no deleterious effect on their survival, we studied the viability of gamma-irradiated fibroblasts over a period of three weeks, as well as their state of proliferation by western blotting. As expected, PCNA labelling was not found in irradiated cells (S2D Fig) whereas their survival rate was constant over a period of three weeks (S2F Fig). Furthermore, we showed that irradiated fibroblasts did not impair the proliferation of $CD49_f^{pos}$ cells, as demonstrated by PCNA expression in a co-culture of irradiated and $CD49_f^{pos}$ cells (S2D Fig). Gamma-irradiated fibroblasts were injected either before (Pre-transplantation) or at the same time (Co-transplantation) as $CD49_f^{pos}$ bMEC. After 10 weeks, mice mammary glands were collected and carmine-stained whole mounts were prepared. As shown in Fig 2, co-transplantation (as well as pre-transplantation) of $CD49_f^{pos}$ bMEC resulted in outgrowth developments (Fig 2A dotted black line and 2B), and this in 50% of the transplantation assays (Fig 2C). No outgrowth was observed when bovine fibroblasts were injected alone (Fig 2C). Obviously, however, no organised epithelial structure was found following transplantation of $CD49_f^{pos}$ cells. Outgrowths corresponded to clusters of cells devoid of any recognisable organisation with stretches of stroma, imbricated in the adipose tissue of the mouse mammary gland (Fig 2B). This behaviour was in striking contrast to what was observed above with bovine explant transplantation.

## Xenograft of candidate bovine mammary epithelial stem cell subpopulations results in tissue outgrowth developments

Previous work in our lab [9, 22] and others [10, 23, 24] highlighted several epithelial cell subpopulations within the bovine mammary gland, including putative MaSC subpopulations and

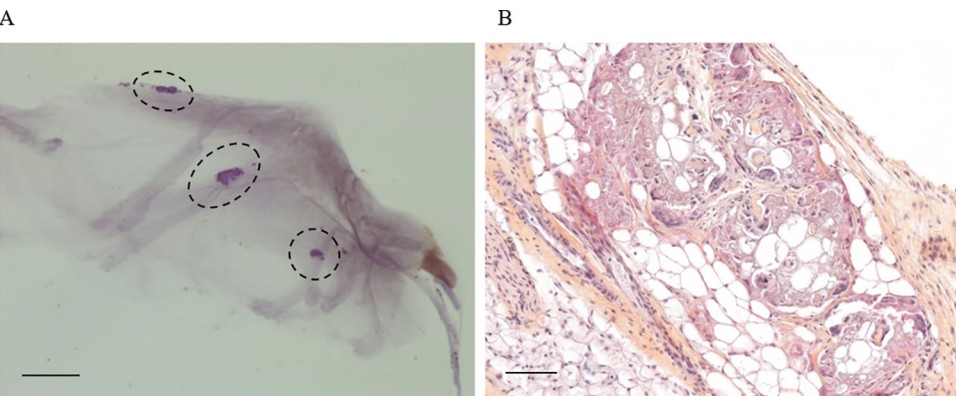

C

| Type of transplant | Control | Co-transplantation | Pre-transplantation |
|---|---|---|---|
| bFIB | + | + | + |
| Sorted bMEC (CD49f$^{pos}$ cells) | - | + | + |
| Frequency of outgrowth development | 0/8 | 4/8 | 4/8 |

**Fig 2. Xenotransplantation of sorted bovine mammary epithelial cells into bovinised murine mammary fat pad results in tissue outgrowth developments.** Irradiated bovine fibroblasts (bFIB) were injected into murine cleared mammary fat pad either without (Control) or with sorted bovine mammary epithelial cells (bMEC = CD49$_f^{pos}$ cells, Co-transplantation), or three weeks before bMEC injection (Pre-transplantation). (A) Representative carmine-stained whole mount of co-transplanted mouse mammary gland. The dotted black lines surround outgrowth developments. Scale bar = 2 mm. (B) Hematoxylin and eosin-stained histological section of a representative outgrowth. Scale bar = 100 μm. (C) Frequency of outgrowth development following transplantation with bFIB, either without (Control) or with sorted bMEC (CD49$_f^{pos}$ cells, Co-transplantation), or with bFIB three weeks before bMEC injection (Pre-transplantation).

their progenies. In many species, MaSC are believed to be contained in a CD49$_f^{high}$CD24$^{pos}$ subpopulation. Other data indicate that CD49$_f^{high}$CD24$^{neg}$ cells are of the basal lineage. Fig 3A displays the gating strategy we used for the sorting of these two-candidate MaSC populations on the basis of the expression of CD49$_f$ and CD24. As shown in Fig 3B transplantation of both cell subpopulations (Co-transplantation with gamma-irradiated bovine fibroblasts) gave rise to 50 to 75% of outgrowth developments. Representative images of outgrowths obtained after transplantation of either CD49$_f^{high}$CD24$^{neg}$ or CD49$_f^{high}$CD24$^{pos}$ cells are shown in Fig 3C left and right, respectively (Fig 3C dotted black line and 3D). As observed above for sorted CD49$_f^{pos}$ mammary epithelial cells, no organised epithelial structure was found after transplantation of these two subpopulations of cells (Fig 3D left and right, respectively for CD49$_f^{high}$CD24$^{neg}$ and CD49$_f^{high}$CD24$^{pos}$ cells). Indeed, for both subpopulations, xenotransplantation resulted in the developments of clumps of cells within the mouse adipose tissue, with stromal stretches containing blood capillaries.

## Discussion

Biological and transplantation studies have led to the conclusion that morphogenesis, homeostasis, and remodelling of the mammary gland epithelium are all dependent upon adult mammary stem cell populations. In the present study, we have established a bovinised transplantation system in order to test and further characterise the putative bovine MaSC

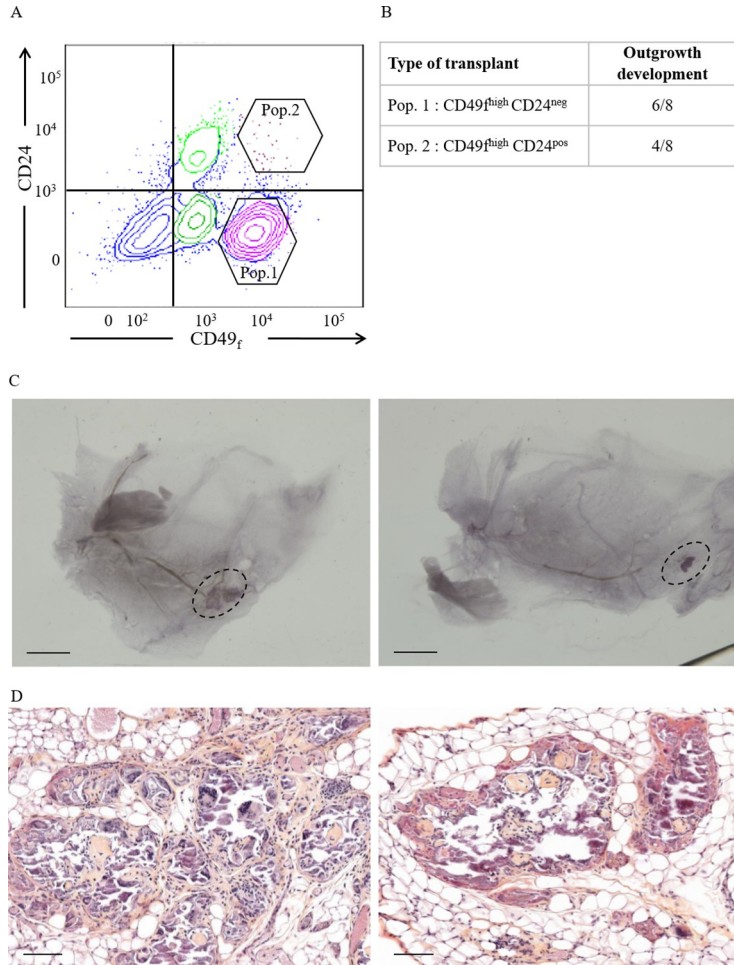

**Fig 3. Xenotransplantation of candidate bovine mammary stem cell populations into bovinised murine cleared mammary fat pad results in outgrowth development.** (A) Dot plot depicting the gating strategy for the sorting of two candidate mammary stem cell populations on the basis of the expression of $CD49_f$ and CD24. Pop. 1: $CD49_f^{high}CD24^{neg}$; Pop. 2: $CD49_f^{high}CD24^{pos}$. (B) Frequency of outgrowth development following transplantation with either Pop. 1 or Pop. 2 with irradiated bovine fibroblasts. (C) Representative carmine-stained whole mount of mouse mammary gland transplanted with either Pop. 1 (left) or Pop. 2 (right) exhibiting outgrowth development (dotted black line). Scale bar = 2 mm. (D) Hematoxylin and eosin-stained histological section of representative outgrowths from mice mammary gland transplanted with either Pop. 1 (left) or Pop. 2 (right). Scale bar = 100 μm.

subpopulations we previously highlighted [1, 9]. To this end, we first tested the transplantation of mammary gland explants into cleared mammary fat pads of immunodeficient female mice. As expected, and as previously demonstrated by others [11, 25–27], implantation of mouse parenchyma explants resulted in the development of the archetypical ductal tree architecture of pubertal mice (see Fig 1A and 1B). We also observed tissue outgrowths in whole mounts of mouse mammary glands after the transplantation mammary gland explants prepared from pubertal heifers. However, in contrast to what was observed following the transplantation of mouse explants, the developing bovine tissue did not invade the entire fat pad. Rather, tissue development remained limited to a modest and compact volume. Similar observations were made by Sheffield and Welsch [13]. Although one cannot totally exclude the possibility that the observed bovine outgrowths partly correspond to the implanted pieces of tissue, we do not favour this hypothesis for the following three main reasons. First, tissue outgrowths were

present in only 75% of the whole mounts, in contrast to 100% when mouse explants were implanted. Second, these tissue outgrowths were larger than the implanted tissue pieces. Third, we observed viable proliferating cells within the epithelial cell layers of the bovine explant xenograft, as indicated by immunofluorescence using the proliferation marker Ki67. These observations suggest that cells of the implanted bovine tissue pieces had proliferated, as was previously demonstrated [24]. This conclusion was supported by the analysis of haematoxylin and eosin-stained histological paraffin sections of the bovine implants which showed tissue morphology similar, if not identical, to that of bovine mammary tissue at the same physiological step [2, 9]. Moreover, immunofluorescence staining of histological sections exhibited identical distribution pattern of the basal marker cytokeratin 14; the luminal marker cytokeratin 7 and the stromal marker collagen type I to that of pubertal bovine mammary gland (see [1]. Altogether, these data also indicate that the mouse mammary fat pad was incompetent in supporting extensive growth of bovine ducts and epithelia. Obviously, the stroma of ruminants is much more fibrous than that of rodents and development of bovine mammary epithelium may therefore require further collagenous fibers. Moreover, it is believed that stromal requirements are species specific [28–30].

Given the above considerations and inspired by the concept of humanization developed for the study of human MaSC, we then decided to develop the bovinisation of the fat pad of recipient mice for transplantation experiments aimed at testing sorted bovine epithelial cell subpopulations. With this aim, we isolated fibroblasts from the mammary subcutaneous adipose tissue of pubertal Holstein heifer and sub-cultured them for several 10th passages before their use in transplantation experiments. These cells quickly showed a remarkable stability, good proliferative capacity with standard doubling time of $\approx 2$ days, and high viability (see the characterisation of bovine fibroblasts in S2 Fig). We also found that irradiated fibroblasts stop proliferating but still survived in vitro. As expected, the injection of irradiated bovine fibroblasts in the mouse mammary fat pad did not lead to any outgrowth. The pre- or co-transplantations of the fibroblasts prior to or together with the sorted cell subpopulations had the same outcome on the frequency of outgrowth developments. These bovinised fat pads served as recipients for transplantation of sorted bMEC ($CD49_f^{pos}$) or candidate bovine mammary epithelial cell stem cell populations ($CD49_f^{high}CD24^{neg}$ or $CD49_f^{high}CD24^{pos}$). We observed a minor reduction of the take rate (50%, 75% at best) following transplantation of these cells, as compared to bovine explants. In all cases, however, no acinar or ductal developments with hollow lumen were observed. This was in striking contrast to what we previously observed with bovine explants. Indeed, outgrowths were made up of clumps of cells surrounded by stretches of stromal fibers and blood capillaries.

The above observations imply that the potential of epithelial cells to regenerate tissue outgrowths following transplantation is held by those expressing high level of $CD49_f$, but not necessarily by CD24 positive cells (see below). This is in agreement with similar studies both in mice [3] and human [8], as well as in a series of experiments aimed at identifying regenerative cellular entities in the epithelium of 7- to 10-month-old Holstein heifers [24]. In this later report, the development of outgrowths was observed upon transplantation of cell subpopulations exhibiting identical phenotypes to our sorted subpopulations ($CD49_f^{high}CD24^{neg}$ and $CD49_f^{high}CD24^{low/med}$), although with different take rates. They observed take rates of 75% with $CD49_f^{high}CD24^{low/med}$ cells and 100% with $CD49_f^{high}CD24^{neg}$, as compared to 50% and 75%, respectively, in our study. On the other hand, Rauner and Barash obtained outgrowths containing multilayered epithelia with hollow lumen [24]. However, one cannot affirm that the cell subpopulations we transplanted here are identical to those tested in their studies. At least two possibilities could be envisioned as to explain these differences. First, it should be noted that the bovine epithelial cell subpopulations transplanted in the Rauner's study were

sorted from mammary parenchyma collected from much younger Holstein heifers (7- to 10-months of age) than those used in the present study (17-months of age). We cannot exclude that cells of the epithelial hierarchy from younger animals (i.e., before or at puberty) are more immature and less oriented than cells from post-pubertal animals, resulting in a more homogeneous MaSC subpopulation after sorting. In line with this concept, the study of the mammary epithelial hierarchy in mice using single cell RNA sequencing has demonstrated a gradual change of cell state and expression pattern throughout mammary gland development (embryonic, foetal, pre- and post-puberty), highlighting a heterogeneity of epithelial cells within the basal and luminal compartments [31]. Functional assays revealed that less than 2% of the adult basal epithelial cells presented mammary stem cell attributes. Nevertheless, no transcriptionally distinct stem cell population could be identified within the basal cell population. Lineage analysis by single cell RNA sequencing revealed the existence of bipotent and long-lived unipotent cells in the murine mammary gland and evidence for a heterogeneous MaSC compartment comprising slow-cycling cells, long- and short-term repopulating cells [32]. These data are corroborated by lineage tracing experiments in vivo that have demonstrated the commitment of multipotent embryonic stem cells in initial mammary gland development. However, it seems that during puberty and the period of adult mammary tissue homeostasis, the expansion and maintenance of each epithelial lineage (myoepithelial or luminal) would instead be due to unipotent lineage-restricted stem cells capable of differentiating into either myoepithelial or luminal lineages [33]. These results highlight the extent of heterogeneity in the phenotypes expressed by each of the luminal and basal cell populations, including MaSC, particularly in late post-natal life. This might explain the difficulties in distinguishing and isolating a population of stem cells, due to the multiple cellular transition states. The heterogeneity of MaSC in the basal population can be illustrated by their phenotypes but also by their differential capacities to generate all the epithelial lineage cells. Indeed, in the Rauner's study, the $CD49_f^{high}CD24^{pos}$ cell population generated both multilayered (basal and luminal committed structures) and monolayered (basal committed structures) outgrowths whereas the $CD49_f^{high}CD24^{neg}$ cells generated mostly monolayers [24]. Importantly, the mammary gland of 17-months heifers obviously underwent several hormonal cycles during puberty that impregnated epithelial tissue with hormones, mainly steroid hormones. This probably affects the undifferentiated epithelial cells like progenitors and MaSC. In conclusion, the use of older heifers, in late puberty, may well have influenced both the heterogeneity of cell state and orientation change of MaSC/progenitor cells explaining, at least in part, the difference in frequency and quality of outgrowth developments obtained in the present transplantation experiments.

The second possibility that may explain the discrepancy between our data and those obtained by the group of Barash is related to the methodology we used for xenotransplantation. A recent study in mice shows that the basal cell population, which is enriched in MaSC, express the epidermal growth factor receptor, presuming a receptivity of these cells to the epidermal growth factor or EGF [34]. EGF may be an activating stimulus for MaSC. However, the processes of MaSC isolation and sorting could be too harsh ending to the damage and/or stripping of the cell surface proteins then altering receptors essential for stem cell functionality. Another possibility resides in the fact that we implanted irradiated fibroblasts. Rather, Rauner and Barash have chosen to co-transplant equal number of non-epithelial cells ($CD49_f^{neg}CD24^{neg}$) with their sorted cell subpopulations. It is therefore quite possible that irradiated fibroblasts alone are not sufficient for fully stimulate stem cells and regenerate mammary tissue.

Further work is therefore needed both to better identify bovine mammary epithelial cells with optimal capabilities of mammary tissue regeneration and to improve our xenotransplantation model to allow these cells to fully express their potential in the neoformation of the normal mammary gland structure that supports the function of milk secretion.

## Supporting information

**S1 Fig. Morphology of virgin murine and bovine mammary glands.** Hematoxylin and Eosin-stained sections of mammary tissue from 8 weeks-old mouse (top panel) and 17-months old heifers (bottom panel) were viewed using a NanoZoomer. Scale bar left panels = 2 mm; scale bar right panels = 100 μm.
(DOCX)

**S2 Fig. Characterisation of the bovine fibroblasts used for the bovinisation of murine cleared mammary fat pad.** Bovine primary fibroblasts were isolated from heifer mammary subcutaneous adipose tissue and sub-cultured during 15 passages before their first utilization in the xenotransplantation assay. (A-B) The clonality of the fibroblastic cell culture was analysed based on the expression of fibroblastic markers and the absence of epithelial markers. (A) Fibroblasts at passage 10 were fixed and analysed by indirect immunofluorescence for the basal epithelial cell marker cytokeratin 14 (green), the luminal epithelial cell marker cytokeratin 7 (orange) and the stromal protein collagen type I (red). Note the absence of epithelial cell markers and the prevalence of the fibroblastic collagen type I marker. Nuclei were counterstained with Hoechst 33342 (blue). Scale bar = 25 μm. (B) A protein fraction was prepared from either cultured bovine fibroblasts (bFIB) at passage 10 or bovine mammary gland parenchyma (PAR) and analysed by SDS-PAGE followed by immunoblotting for either the fibroblastic markers vimentin (VIM; Mr. 57 kDa) and smooth muscle actin alpha (αSMA; Mr. 42 kDa) or the epithelial cell markers E-cadherin (CDH1; Mr. 120 kDa) and cytokeratin 19 (KRT 19; Mr. 44 kDa). Molecular mass markers (kDa) are shown on the left. (C) Cultured bovine fibroblasts at passage 10 were fixed and subjected to immunofluorescence using antisera against telomerase (red) and vimentin (green). The telomeric activity is found in fibroblasts nuclei, maintaining their proliferation capacity through passages. Nuclei were counterstained with Hoechst 33342 (blue). Scale bar = 25 μm. (D) Protein fractions were prepared from irradiated bFIB cultured for the indicated days with or without sorted $CD49_f^+$ epithelial cells and analysed by SDS-PAGE followed by immunoblotting for PCNA. (E) Doubling time and viability of the cultured bovine fibroblasts at passages 5, 10, 15 and 20. (F) Viability of cultured irradiated bovine fibroblasts over a three-week period.
(DOCX)

**S1 Table. Antibodies used for flow cytometry (FACS), western blotting and immunofluorescence analyses.**
(DOCX)

**S1 Graphical abstract.**
(TIF)

## Acknowledgments

The authors are grateful to the staff of the dairy experimental farm IEPL (INRAE, UMR PEGASE, Le Rheu, France) and the Infectiology of fishes and rodent facility (INRAE, UE 0907, Jouy-en-Josas, France; doi.org/10.15454/1.5572427140471238E12 Infectiology of fishes and rodent facility) for animal care. We thank Laurent Deleurme and Florence Boutillon from the CytomeTRI and ARCHE platforms of BIOSIT (Rennes, France), respectively, for technical assistance. This work greatly benefited from the facilities of the @BRIDGE platform (INRAE, Université Paris-Saclay, Jouy-en-Josas, France) for histological sections and we thank Julie Rivière and Marthe Vilotte for their assistance.

## Author Contributions

**Conceptualization:** Laurence Finot, Cathy Hue-Beauvais, Fabienne Le Provost, Eric Chanat.

**Data curation:** Laurence Finot, Cathy Hue-Beauvais, Etienne Aujean.

**Formal analysis:** Laurence Finot, Cathy Hue-Beauvais, Etienne Aujean.

**Investigation:** Laurence Finot, Cathy Hue-Beauvais, Etienne Aujean, Fabienne Le Provost, Eric Chanat.

**Methodology:** Laurence Finot, Cathy Hue-Beauvais, Etienne Aujean, Fabienne Le Provost.

**Resources:** Cathy Hue-Beauvais.

**Supervision:** Fabienne Le Provost, Eric Chanat.

**Validation:** Fabienne Le Provost, Eric Chanat.

**Visualization:** Laurence Finot.

**Writing – original draft:** Laurence Finot, Eric Chanat.

**Writing – review & editing:** Cathy Hue-Beauvais, Etienne Aujean, Fabienne Le Provost.

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
