## [Decision Letter · Decision Letter 0]

13 Feb 2024

PONE-D-23-42849Sorted stem/progenitor epithelial cells of pubertal bovine mammary gland present limited potential to reconstitute an organised mammary epithelium after transplantationPLOS ONE

Dear Dr. Chanat,

Thank you for submitting your manuscript to PLOS ONE. After careful consideration, we feel that it has merit but does not fully meet PLOS ONE’s publication criteria as it currently stands. Therefore, we invite you to submit a revised version of the manuscript that addresses the points raised during the review process.

We look forward to receiving your revised manuscript.

Kind regards,

Heiner Niemann

Academic Editor

PLOS ONE

Reviewers' comments:

Reviewer's Responses to Questions

**Comments to the Author**

1. Is the manuscript technically sound, and do the data support the conclusions?

Reviewer #1: Partly

Reviewer #2: Partly

2. Has the statistical analysis been performed appropriately and rigorously? 

Reviewer #1: N/A

Reviewer #2: N/A

3. Have the authors made all data underlying the findings in their manuscript fully available?

Reviewer #1: Yes

Reviewer #2: Yes

4. Is the manuscript presented in an intelligible fashion and written in standard English?

Reviewer #1: Yes

Reviewer #2: Yes

5. Review Comments to the Author

Reviewer #1: The paper by Finot and colleagues is timely as it illustrates the complexity of post-natal organogenesis / organoid research. As such there is much to like about the work. The authors performed a interesting series of experiments, however in this reviewers opinion the work has too many open ends that need to be addressed in order to make the observations publicible. In fact, most of the aspects brought forward in the discussion should have been addressed. For example: concerning the explants: were the tissue responses (i.e. angiogenesis) similar for murine and bovine pads. Are inactivated fibroblasts able to switch phenotype, what is the origin of mammary gland mesenchymal cells, EMT?? There potential to produce growth factors and cytokines should be tested. A further aspect brought forward in the discussion and that is not tested are the contrasts with "Rauner's" study. What is the hormonal status of your mouse stain.

Reviewer #2: In this manuscript, Finot et al. tested the developmental potential of bovine stem/progenitor epithelial cells enriched from the mammary gland. The putative mammary stem cells (MaSC)/progenitor cells were characterized according to the expression of two surface markers CD49f and CD24. The idea of attempting to transplant enriched mammary stem cells and progenitor cells into cleared fat pads of female mice is very interesting, however, the capability of these cell populations to generate mammary gland structure seems to be too low. The enriched putative population of MaSC in bovine is low and represents only 3.3% of total mammary cells at puberty and is decreasing consistently to lactation (Finot et al., 2019).

Comments

The author needs to improve the quality of the Figures, especially tables that are not good enough for publication.

Line 36 and Line 585: Could you please explain how you confirmed the self-renewing potential of putative mammary epithelial stem cells transplanted to the bovine mammary gland?

Line 37-38: Please rewrite this sentence: “Sorted cells, however….”

Line 57 and Line 118: Please remove “Doing so,..”

Line 131-133: Please add here more detail which subpopulations were used for xenotransplantation (mammary epithelial cells (CD49fpos), bovine mammary epithelial stem cells (CD49fhigh/CD24pos) and CD49fhigh/CD24neg). Could you please add also more information about the sorted cell populations in Material & Methods or Results, especially the differences between enrichment of CD49fpos and CD49fhigh/CD24neg cell subpopulations used for transplantation, and include information about the purity of sorted cell populations.

Line 170: For the explant xenograft cryo-conserved bovine mammary tissues were used. It is not clear if mouse explants were also used after the freezing-thawing procedure or fresh. Can freezing influence the survival and potential of the cells? For cell transplantation (Line 179) sorted cells were isolated from frozen tissue and thereafter frozen again. Is not possible to dissociate cells from fresh tissue, sort them, and then freeze them to avoid two-time freezing? Each thawing and freezing process could impair stem cells.

Line 197: For the cell transplantation 15,000 bMEC and 5,000 putative stem/progenitor cells or/and 1x105 inactivated fibroblast were used. Did you test before the cell number that will be optimal for successful cell transplantation? In other stem cell studies to prove the regenerative potential of progenitor/stem cells millions of cells need to be used.

Line 320: The authors claim that xenografts of the heifer mammary gland explant “regenerate” a bovine-like mammary epithelium. The transplanted explant survived in a mouse fat pad and can be used as a positive control for testing xenotransplantation but for me is not clear how the regenerative potential was confirmed. Could you please explain it? The proliferating Ki67-positive cells represent a very low number (Figure 1C). Could you please perform co-staining of these cells with other markers to confirm the cell type and define the number of Ki67-positive cells? It would be helpful to see similar staining presented in Figure 1 in bovine tissue after thawing and before transplantation. It would be also interesting to stain the explants with CD49f and CD24 markers to localize and confirm the survival/proliferation of stem cells/progenitor cells.

Line 362-428: Xenograft of candidate bovine mammary epithelial and epithelial stem cell subpopulations results in tissue outgrowth developments. To better confirm the multipotential of the transplanted cells additional cell type-specific staining in bovine outgrowths will be important.

The discussion part is too long.

6. PLOS authors have the option to publish the peer review history of their article (what does this mean?). If published, this will include your full peer review and any attached files.

Reviewer #1: No

Reviewer #2: No

---

## [Author Response · Author response to Decision Letter 0]

19 Apr 2024

Dear Heiner Niemann,

Thank you for your mail of February 13, 2024 and the comments of the two Reviewers. We are very pleased about the helpful review and the Reviewers' constructive criticisms and recommendations.

We have re-submitted a revised version of the manuscript in which most suggestions of the Reviewers have been incorporated. A point-by-point response to the Reviewers' comments could be found below.

We hope that this new version will find your approval.

With best regards.

Response to Journal requirements.

Point #4:

We apologize for having used “data not shown” twice in the first version of our manuscript. This is obviously in contradiction with the concept of data sharing.

Concerning the first occurrence, we added the data to the supplementary figure, panel D (see Supplementary Figure S1D). The figure and the figure legend have been modified accordingly.

In the second case, we simply removed the data since those are not a core part of the present work. These data, however, would be part of a separate study.

The Results and Discussion sections have been amended accordingly.

Point #5:

Original blot/gel images as well as all other images and data could be found at the following repository URL:

https://entrepot.recherche.data.gouv.fr/dataset.xhtml?persistentId=doi:10.57745/NQZKHI&version=1.0

https://doi.org/10.57745/NQZKHI

Response to the reviewer's comments.

Reviewer 1:

Response to General comments:

* Line number refers to MS revised with marks.

“the work has too many open ends that need to be addressed in order to make the observations publicible.”

Obviously, we would have love to improve the impact of our current research and be able to answer some of the important questions we raised in the Discussion section. It is clear, however, that most, if not all, of these important points constitute large projects in themselves and would be the subject of separate studies. In our opinion, such data would therefore be published as additional papers.

Response to specific points:

1- “…concerning the explants: were the tissue responses (i.e. angiogenesis) similar for murine and bovine pads….?

As the reviewer will be able to judge for himself in wide field images from H&E stained samples, blood vessels were also found throughout the mice fat pad when bovine explant were transplanted (see Fig R1-1 appended below). The development of the venous endothelium was also observed in mice grafted with murine mammary gland explant, as shown by alphaSMA labelling in Fig R1-2. AlphaSMA nicely decorated endothelial cells (thick arrows), but also labelled myoepithelial cells at the basal side of the mammary epithelium (thin arrows).

We could add these data if requested by the reviewer.

2- “Are inactivated fibroblasts able to switch phenotype, what is the origin of mammary gland mesenchymal cells, EMT??”

It should be noted that fibroblasts were isolated from the mammary subcutaneous tissue. This is specified in the Methods section. We collected samples from the white matter, outside the mammary parenchyma, to avoid contamination by epithelial tissue.

“There potential to produce growth factors and cytokines should be tested.”

We agree and have therefore tested the expression of a panel of growth factors, receptors, extracellular matrix proteins and cytokines by qPCR. Moreover, we have analysed both standard and irradiated bovine fibroblasts (see Fig R1-3). Results are twofold: all the expected growth factors and receptors were found to be expressed in the in vitro cultured bovine fibroblasts and we show that irradiation had few effects except for substantial increases in FGF1 and EGF expression. Concerning the immune and oxidative responses, the expression of the tested genes was low as indicated by the amplification curves. However, irradiation was found to have a drastic effect on the level of expression of the pro-inflammatory cytokine IL1beta. Such effect has previously been reported (DOI:10.3389/fcell.2021.539893). However, many transplantation studies have used this treatment without impairing tissue development (xeno or allograft).

These data strongly suggest that irradiation does not induce a “switch” in fibroblasts phenotype, at least in vitro. However, we cannot firmly exclude that transplanted irradiated fibroblasts present a different fate; e.g. they could undergo a mesenchymal-epithelial transition and therefore no longer provide the awaited stimuli for epithelial development. We believe it would be very difficult to test this in vivo. Finally, it should be noted that we observed no development when transplanting irradiated fibroblasts. 

These qPCR data could be added to the supplementary figure if requested by the reviewer.

3- “A further aspect brought forward in the discussion and that is not tested are the contrasts with "Rauner's" study.”

The differences observed between our study and Rauner's work may be due to multiple reasons. Understanding the causes of these differences is therefore particularly difficult.

First, recipient mice are of different strains (BALB/c vs. NOD-SCID). Second, “bovinisation” was following a quite different approach, with 100% irradiated bovine fibroblasts isolated from the mammary subcutaneous tissue. Rauner co-grafted “non-epithelial” cells, that is to say a pool of cells remaining following the sorting of epithelial cells and which are therefore unidentified. In addition, they used Matrigel, a murine extracellular matrix. In contrast, we wanted to provide a bovine extracellular matrix. Finally, as discussed, bovine mammary gland samples were from 17 months old cows instead of 7-10 months in the Rauner’s study, with the possibility that stem cells from post-pubertal animals would be of different “quality”, being more mature. Testing all these parameters would be a tremendous work and, in our opinion, beyond the present study.

4- “What is the hormonal status of your mouse stain.”?

Concerning the last point of this Reviewer’s criticism, namely what is the hormonal status of the mouse strain involved in the present study, it must first be clarified that we used young virgin mice which were not yet adults at the time of transplantation. Indeed, the mice were 3 weeks old at the time of fat pad clearing and transplantation. In the case of bovine fibroblast pre-transplantation, sorted cells were graft when mice were six-week-old. On the other hand, the mice were adults at the time of the xenograft analysis carried out 10 weeks later on still virgin animals. These facts are described in the Methods section but could be emphasized if required by the reviewer.

Also note that we have not attempted to mate the mice because this strain is known for its breeding difficulties. We also did not attempt to implant a hormonal patch to promote the development of breast tissue from our grafts. The xenograft developments were therefore studied under the hormonal conditions that normally occur during puberty in this strain, as well as in young adult, with several oestrous cycles between 6 and 10 weeks. Moreover, our data showed that these conditions allow the normal development of homotypic mammary tissue grafts (see Fig 2, panel A, top image).

Reviewer 2:

“The author needs to improve the quality of the Figures, especially tables that are not good enough for publication.”

As suggested, we have modified Tables in all figures.

“Line 36 and Line 585: Could you please explain how you confirmed the self-renewing potential of putative mammary epithelial stem cells transplanted to the bovine mammary gland?” Line 37-38: Please rewrite this sentence: “Sorted cells, however….”

This comment is apparently due to a misunderstanding of the reviewer because we probably did not describe our data correctly. Indeed, it seems that the reviewer understood that the capacity of transplanted cells to self-renew was demonstrated in the present report. In fact, it is undoubtedly necessary at this stage to clarify that the self-renewal capacity of the cells studied in this report was actually determined in a previous study by our group (see Finot L, Chanat E, Dessauge F. Molecular signature of the putative stem/progenitor cells committed to the development of the bovine mammary gland at puberty. Sci Rep. 2018;8(1):16194. doi: 10.1038/s41598-018-34691-2.). What we meant here is that the present data are in agreement with this notion of cell self-renewing of the transplanted sorted cells.

This being apparently misleading, we decided to delete the sentences from the abstract (see line 36) and last § of the discussion (line 585 now line 593).

As suggested, we also modified the sentence “Sorted cells, however….” as follows:

“In conclusion, sorted cells showed …..”

Line 57 and Line 118: Please remove “Doing so,..”

We modified both sentences as suggested by the reviewer.

Line 131-133: Please add here more detail which subpopulations were used for xenotransplantation (mammary epithelial cells (CD49fpos), bovine mammary epithelial stem cells (CD49fhigh/CD24pos) and CD49fhigh/CD24neg). Could you please add also more information about the sorted cell populations in Material & Methods or Results, especially the differences between enrichment of CD49fpos and CD49fhigh/CD24neg cell subpopulations used for transplantation, and include information about the purity of sorted cell populations.

A good point. As suggested, we have re-written the sentence line 131-133 to incorporate more details about the transplanted bovine cell subpopulations.

The Materials and Methods section has also been amended as required by the reviewer (see line 189-194 and 200-202).

Line 170: For the explant xenograft cryo-conserved bovine mammary tissues were used. It is not clear if mouse explants were also used after the freezing-thawing procedure or fresh. Can freezing influence the survival and potential of the cells? For cell transplantation (Line 179) sorted cells were isolated from frozen tissue and thereafter frozen again. Is not possible to dissociate cells from fresh tissue, sort them, and then freeze them to avoid two-time freezing? Each thawing and freezing process could impair stem cells.

To respond to the first part of the reviewer query, we have precised in the relevant Materials section that mouse explant for transplantation were prepared extemporaneously.

Concerning the freezing of bovine sorted cells, the viability of these cells was tested before injection and was found to be greater than 90%. Furthermore, we showed in Figure S1, panel F, that CD49fpos cells co-cultured with irradiated bovine fibroblasts proliferated using Western blot and PCNA labelling. We also tested that each sorted cell subpopulation proliferated in culture. Of course, we cannot exclude that freezing’s may have an impact on the survival and potential of cells. However, the fact that both the slaughter house and the transplantation facility were far from our sorting platform led us to cryo-preserve cell samples. We have also chosen to cryo-conserve the sorted cells to limit possible phenotypic changes before injection.

Line 197: For the cell transplantation 15,000 bMEC and 5,000 putative stem/progenitor cells or/and 1x105 inactivated fibroblast were used. Did you test before the cell number that will be optimal for successful cell transplantation? In other stem cell studies to prove the regenerative potential of progenitor/stem cells millions of cells need to be used.

We did not test the optimal number of cells for transplantation and designed our procedure based on the literature on bovine cell xenotransplantation. We were inspired specifically by the work of Martignani, but also by the approach used in Rauner’s study. In both reports, the number of cells injected was in the thousands rather than millions. It was also a matter of reducing the number of tests and controls, which were already numerous, and of reducing the number of mice sacrificed during this experimentation.

Line 320: The authors claim that xenografts of the heifer mammary gland explant “regenerate” a bovine-like mammary epithelium. The transplanted explant survived in a mouse fat pad and can be used as a positive control for testing xenotransplantation but for me is not clear how the regenerative potential was confirmed. Could you please explain it? The proliferating Ki67-positive cells represent a very low number (Figure 1C). Could you please perform co-staining of these cells with other markers to confirm the cell type and define the number of Ki67-positive cells? It would be helpful to see similar staining presented in Figure 1 in bovine tissue after thawing and before transplantation. It would be also interesting to stain the explants with CD49f and CD24 markers to localize and confirm the survival/proliferation of stem cells/progenitor cells.

Several arguments in favour of obtaining development of bovine explants and their regenerative potential were developed in the Discussion section (see lines 468-472), including KI67 labelling. As suggested by the reviewer, we have now quantified this labelling within the grafted bovine explant, as well as explants before transplantation. Furthermore, we decided to also quantify this labelling in transplanted murine mammary gland explants for comparison. We found that the proliferation rate decreased from ≈ 4,8 % in the bovine explant to ≈ 2,8% in the grafted explant. The proliferation rate was similar in the grafted murine explant. These data have been added to Figure 1, panel E, and the figure legend, Methods and Results sections have been modified accordingly. These proliferation rates might appear low but this could be explained by the fact that here we analyse mammary glands from virgin adult animals, i.e. at a period when mammary development has yet occurred and is no longer intense. Consistent with these data, a previous study from our group showed that the proliferation rate in the developing mammary gland of goats at mid-gestation was ≈ 4% (Panzuti, C. et al. 2019; DOI: 10.1017/S0022029919000505).

As shown in Figure 1, panel C, KI67 labelling is localised throughout the developed mammary structures and obviously labelled various cell types. How then to choose a relevant marker to type non-epithelial cells? However, the majority of the labelling was detected within the epithelial structures. In our opinion, co-staining would therefore not be more informative.

Regarding the last point of the reviewer, we believe that this approach will not allow the identification of the stem cells since all epithelial cells express CD49f and both stem and luminal progenitor cells express CD24. In addition, discrimination of these cells based on CD49f expression level, indeed possible by FACS, will be technically difficult by immunofluorescence. As shown in Figure 1 of Rauner & Barash 2012 (DOI: 10.1371/journal.pone.0030113), CD49f and CD24 co-staining labelled all epithelial cells, basal and luminal, failing to identify mammary stem cells. Moreover, stem cells are very few in number in the tissue niches, as you know.

Line 362-428: Xenograft of candidate bovine mammary epithelial and epithelial stem cell subpopulations results in tissue outgrowth developments. To better confirm the multipotential of the transplanted cells additional cell type-specific staining in bovine outgrowths will be important.

We completely agree that it would have been important to further characterise the bovine outgrowths. In our case, however, the outgrowths did not exhibit the typical epithelial organization of the mammary gland tissue. It is therefore not certain that the epithelial markers would have shown their precise localisation, or even that they would be expressed. Additionally, we must admit that unfortunately all samples were processed for H&E staining and it is obviously not possible to redo the entirety of this experiment in the time of this review.

The discussion part is too long.

The Discussion section has been substantially shortened.

For appended Figures, see downloaded file "Finot et al PLOS ONE 2024 response to Reviewers"

---

## [Decision Letter · Decision Letter 1]

9 Aug 2024

PONE-D-23-42849R1Sorted stem/progenitor epithelial cells of pubertal bovine mammary gland present limited potential to reconstitute an organised mammary epithelium after transplantationPLOS ONE

Dear Dr. Chanat,

Thank you for submitting your manuscript to PLOS ONE. After careful consideration, we feel that it has merit but does not fully meet PLOS ONE’s publication criteria as it currently stands. Therefore, we invite you to submit a revised version of the manuscript that addresses the points raised during the review process.

We look forward to receiving your revised manuscript.

Kind regards,

Yi Li, Ph.D.

Academic Editor

PLOS ONE

Journal Requirements:

Reviewers' comments:

Reviewer's Responses to Questions

**Comments to the Author**

1. If the authors have adequately addressed your comments raised in a previous round of review and you feel that this manuscript is now acceptable for publication, you may indicate that here to bypass the “Comments to the Author” section, enter your conflict of interest statement in the “Confidential to Editor” section, and submit your "Accept" recommendation.

Reviewer #2: All comments have been addressed

Reviewer #3: (No Response)

2. Is the manuscript technically sound, and do the data support the conclusions?

Reviewer #2: Yes

Reviewer #3: Partly

3. Has the statistical analysis been performed appropriately and rigorously? 

Reviewer #2: Yes

Reviewer #3: N/A

4. Have the authors made all data underlying the findings in their manuscript fully available?

Reviewer #2: Yes

Reviewer #3: Yes

5. Is the manuscript presented in an intelligible fashion and written in standard English?

Reviewer #2: Yes

Reviewer #3: No

6. Review Comments to the Author

**Reviewer #2: **The authors have satisfactory reply to most of my comments and concerns. I suggest to accept this paper for publication in the actual form.

**Reviewer #3:** Mammary gland stem cells and progenitor cells are crucial in mammogenesis, and the understanding of these cell populations in bovines is important for the study of milking cows. In this manuscript, the authors studied the potential of certain mammary epithelial populations in reconstituting bovine mammary epithelium in mice. By transplanting bovine mammary gland pieces into cleared mouse mammary fat pad, the authors observed outgrowth of the transplanted pieces with organized epithelial structures. On the other hand, sorted bovine mammary epithelial cells expressing CD49f (supplemented by a bovine fibroblast cell line) only led to outgrowth, but not the epithelial structures. This suggests the limited stemness of the sorted populations. Certain figures in the manuscript need some additional data to better support part of the conclusions. The manuscript may also benefit from some language editing. In summary, this manuscript still needs some improvement before it is published.

Specific comments:

1. In Fig 1, the authors showed the H&E and immunofluorescence staining of the outgrowths generated by xenografted bovine mammary gland explants to demonstrate the structure, and concluded that the outgrowths are highly reminiscent of bovine mammary gland epithelium. To better support this conclusion, a bovine mammary gland control should be included to show the resemblance, and possibly a mouse mammary gland control as well to show the difference.

2. In Figs 2&3, the authors showed the H&E staining to demonstrate outgrowths without organized epithelial structure generated by sorted bovine epithelial cells. The authors may want to do some immunostaining to confirm the outgrowths are indeed composed of epithelial cells.

3. Fig 1B and the middle panel of Fig 1C do not seem very clear when they are magnified. The authors may want to further improve the quality of the pictures.

4. In Fig 2C, the authors included “fibroblast only” as control. It might be of interest to include a “no fibroblast” control as well.

5. The language of the manuscript may need some improvement.

6. There are some grammar issues and typos in the manuscript. For example, in line 63 “as a results” should be “as a result”, and in line 403 “we though that” probably should be “we thought that”.

7. PLOS authors have the option to publish the peer review history of their article (what does this mean?). If published, this will include your full peer review and any attached files.

Reviewer #2: No

Reviewer #3: No

---

## [Author Response · Author response to Decision Letter 1]

17 Sep 2024

Dear Dr. Yi Li,

Following your mail of August 8, 2024 and the helpful comments of Reviewer #3, we are very pleased to re-submitt a revised version of the manuscript in which most suggestions of the Reviewer have been incorporated.

A point-by-point response to the Reviewers' comments could be found below.

We hope that this last version of our manuscript will find your approval.

With best regards.

Response to the reviewer's comments.

Response to General comments:

“Certain figures in the manuscript need some additional data to better support part of the conclusions. The manuscript may also benefit from some language editing. »

As suggested, we have improved images in Fig. 1 and have added a Figure in the Supplementary materials. Regarding the second point, we carried out a thorough rereading of the manuscript and corrected it.

Response to specific points:

“1. In Fig 1, the authors showed the H&E and immunofluorescence staining of the outgrowths generated by xenografted bovine mammary gland explants to demonstrate the structure, and concluded that the outgrowths are highly reminiscent of bovine mammary gland epithelium. To better support this conclusion, a bovine mammary gland control should be included to show the resemblance, and possibly a mouse mammary gland control as well to show the difference. »

We agree and have therefore added images of H&E-stained virgin mouse and cow mammary gland tissues as supplementary Fig.1. The Results and “Materials and Methods” sections have been amended accordingly. Original images were added to the redepository URL.

“2. In Figs 2&3, the authors showed the H&E staining to demonstrate outgrowths without organized epithelial structure generated by sorted bovine epithelial cells. The authors may want to do some immunostaining to confirm the outgrowths are indeed composed of epithelial cells.”

We fully agree that it would have been important to characterise the bovine outgrowths in more detail in order to prove that they are at least partly composed of epithelial cells. Unfortunately, all samples were processed for H&E staining and it is obviously not possible to repeat all these experiments in the time of this review. However, we would like to emphasize that it is doubtful whether the epithelial markers would have exhibited their precise localisation, or even whether they would have been expressed since the outgrowths did not display the typical epithelial organization of mammary gland tissue.

“3. Fig 1B and the middle panel of Fig 1C do not seem very clear when they are magnified. The authors may want to further improve the quality of the pictures.”

As requested, we have improved the quality of Figure 1, notably making the resolution of the images in Fig.1C uniform, with all panels now set to 300 DPI.

“4. In Fig 2C, the authors included “fibroblast only” as control. It might be of interest to include a “no fibroblast” control as well.”

We did not test the transplantation in the absence of bovine fibroblast. The Reviewer is right that this might have been a standard control to do. However, we designed our procedure based on the literature on bovine epithelial cell xenotransplantation in which “bovinisation” of the mouse mammary gland appears to be a prerequisite for successful transplant development. We therefore omitted this control. It was also a matter of reducing the number of tests and controls, which were already numerous, and of reducing the number of mice sacrificed during this experimentation.

“5. The language of the manuscript may need some improvement. & 6. There are some grammar issues and typos in the manuscript. For example, in line 63 “as a results” should be “as a result”, and in line 403 “we though that” probably should be “we thought that”.”

The whole MS was thoroughly edited. We corrected the indicated mistakes as well as some others.

---

## [Editor Report · Decision Letter 2]

29 Sep 2024

Sorted stem/progenitor epithelial cells of pubertal bovine mammary gland present limited potential to reconstitute an organised mammary epithelium after transplantation

PONE-D-23-42849R2

Dear Dr. Chanat,

We’re pleased to inform you that your manuscript has been judged scientifically suitable for publication and will be formally accepted for publication once it meets all outstanding technical requirements.

Kind regards,

Yi Li, Ph.D.

Academic Editor

PLOS ONE
---

## [Editor Report · Acceptance letter]

9 Oct 2024

PONE-D-23-42849R2 

PLOS ONE

Dear Dr. Chanat, 

I'm pleased to inform you that your manuscript has been deemed suitable for publication in PLOS ONE. Congratulations! Your manuscript is now being handed over to our production team.

Kind regards, 

on behalf of

Dr. Yi Li 

Academic Editor

PLOS ONE